# Developing a machine learning model to map new-build gentrification: A mixed-methods approach

Maya Mueller[1]*, Isaac Quaye[2], Shengao Yi[3], James Foley[2], Reeya Shah[2], Xiaojiang Li[3‡], Hamil Pearsall[2‡], Simi Hoque[1‡]

**1** Department of Civil, Environmental, and Architectural Engineering, Drexel University, Pennsylvania, Philadelphia, United States of America, **2** Department of Geography, Environment and Urban Studies, Temple University, Pennsylvania, Philadelphia, United States of America, **3** Department of City and Regional Planning, University of Pennsylvania, Pennsylvania, Philadelphia, United States of America

☉ These authors contributed equally to this work.
‡ These authors also contributed equally to this work.
* mm5446@drexel.edu

## Abstract

New-build gentrification, a type of gentrification which is connected to newly built development, has radically transformed the appearance of neighborhoods across the United States. However, the literature is lacking discussion on the built component of the new-build gentrification process, which can lead to inaccurate maps and projections of gentrification trends. Recent advancements in machine learning (ML), specifically computer vision models that apply neural network "deep mapping" algorithms, have found application in the research for their ability to track changes in urban streetscapes. In our research, we trained machine learning models to identify new-build development with architectural traits that reflect visual cues of gentrification according to local residents. With Philadelphia as our study area, we drew on the insight of community-based focus groups to identify characteristics that denote new-build gentrification for the city. We compared our audit of new-build gentrification development with municipal permit License and Inspections (L&I) data, using Kernel Density Estimate (KDE) maps to visualize the spatial trends of both datasets. Our final fine-tuned ResNet-50 model achieved an 84.0% test accuracy and an 84.0% Area Under the Curve (AUC) score. Our research contributes a novel mixed-methods approach that integrates community input with Artificial Intelligence (AI) to identify locally-specific gentrification traits.

## Introduction

Gentrification occurs when a neighborhood is restructured to accommodate a wealthier social class and results in rapid changes to the urban landscape [1–3]. The gentrification process alters the built environment in all cases, but the literature lacks

**Data availability statement:** All relevant data are within the paper and its Supporting Information files and in provided with an open access DOI to a GitHub where the minimal dataset and code is made available (github.com/niiquaye70/Mapping-New-Build-Gentrification-with-Machine-Learning/tree/main/data).

**Funding:** This research was supported through a grant from the National Science Foundation Award #2312047 (awarded to SH) and #2312048 (awarded to HP). The url to the funder website is https://www.nsf.gov/ for both awards. The full name of both funders of both awards is the organization of the National Science Foundation. The sponsors and funders did not play a role in the study design, data collection, analysis, decision to publish, nor the preparation of the manuscript.

**Competing interests:** The authors have declared that no competing interests exist.

sufficient discussion on what types of built changes connect with gentrification for different localities.

Machine learning-based computer vision models have recently found utility in gentrification research for their ability to map changes to property frontages [4,5]. These models draw on historical Street View Imagery (SVI), such as Google Street View (GSV) that provides a back catalogue of panoramic images from a pedestrian's point of view dating back to 2005. SVI-based audits demonstrate the validity and inter-reliability necessary to be a valid surrogate for in-person field surveys [6–8]. In sum, SVI on its own provides a more accessible, less intensive way to perform field surveys that were traditionally done in-person. SVI in conjunction with ML allows researchers to audit large regions and automate the auditing process itself [9].

Access to SVI and ML modeling techniques pose a solution to developing a system for measuring changes in the built environment, but there is no precedent in the literature for a systematic, replicable methodology to measure the built traits that indicate gentrification occurrence [10]. Audits vary in their operationalizing of gentrification, with some studies connecting gentrification with the privatization of space and exclusivity of certain built design [11–13] or with landscape beautification and the lack of graffiti and trash [14]. Although prolonged disinvestment can be associated with urban decay, this operationalization may overlook the visual variety of neighborhoods. In Philadelphia, for example, graffiti and street art can be a form of beautification, and visible trash exists even in wealthier, gentrifying neighborhoods. Oftentimes residents and communities find ways to beautify their neighborhoods with murals, community gardens, and self-funded improvements, and upkeep the integrity of their buildings without external capital investment.

As we can see, audits of gentrification's built changes may not necessarily reflect these local variations, nor the ways in which the nature of gentrification's impact on the built landscape can change over time. ML-based audits often turn to prior literature [specifically, that of 4, 8] in the labeling of data to train their models. These ML-based, perceptual image recognition models are black-box in that users cannot trace model judgements to a specific image feature. Even high performing models may use visual shortcuts to produce accurate predictions as they lack any built-in knowledge of semantic objects (e.g., car, person, sky). Coupled with the black-box nature of these algorithms, the lack of clarity and transparency behind how the training data images are selected can result in added uncertainty over whether these models are actually honing in on the desired gentrification indicators.

As ML advancements continue to open doors for studying built changes such as gentrification, it is imperative to understand the extent of regional differences and ensure that there is consensus between researchers and communities that experience gentrification firsthand. Moreover, with increased transparency in our methods development, we can evaluate model performance based on pre-defined objectives (e.g., the ability to identify a certain architectural feature) and moreover mitigate the chance that the model learns implicit social biases with respect to perceptions of what gentrification, poverty, investment, and disinvestment look like [15–17]. For

these reasons, the current study qualifies the methods with which we produce our data inputs and is explicit in how we, as researchers, shape our perceptions of gentrification.

In our study, we draw on the insight of focus groups across Philadelphia to identify characteristics that describe gentrification in their respective neighborhoods. We focus on new-build gentrification, a type of gentrification connected to new-build residential development [1]. Whereas classic gentrification is often led by laypeople that rehabilitate historic housing that they themselves inhabit, new-build gentrification operates on a larger scale and is developer-led. Rather than operating in a competitive market, these developers represent the elite minority of suppliers that have the capital to flip entire neighborhoods and deploy marketing strategies to commodify a certain type of urban lifestyle to potential buyers [18–23]. New-build gentrification tends to be more visually distinct in architectural style and in the scale of built changes, ideal for a computer vision model that may struggle to pick up on subtle renovations.

With a diagnostic for identifying gentrification, we ran the ML model ResNet-50 to test ML's capacity in accurately categorizing buildings according to these criteria. Unlike prior research that apply deep mapping models to map classic gentrification and assess the utility of deep mapping as a replacement for permit data [4,5], our research focuses on developing and implementing a methodology for a ML model that identifies new-build gentrification according to protocol that is dictated by the qualitative, community-based component of the research. As a secondary aim, we test whether deep mapping models can identify regionally specific trends of gentrification in Philadelphia that are identified in our focus groups. In this way, the current paper opens discourse on how these increasingly technical quantitative tools can be connected back to a more ethnographic, lived component of gentrification. Whereas deep mapping allows us to sift through large quantities of historical data on landscape change, qualitative perspectives ground the research by giving insight to the visual complexity and local variation of gentrification changes.

To the best of our knowledge, this is the first gentrification study to draw on community input to develop a ML computer vision model, the first to demonstrate that ML can learn regionally specific architectural traits that are culturally understood to be gentrification-specific, and is also the first to apply gentrification-based computer vision models on a highly compact city like Philadelphia with its characteristic attached rowhomes and dense development patterns.

## A review of machine learning for gentrification

Machine learning (ML) has been utilized to model and predict gentrification occurrence and is a promising alternative to non-ML regression modeling in its capacity to draw meaningful patterns from large Census datasets and produce accurate predictions despite data gaps and irregularities [15,24,25]. ML is a relatively new method in gentrification research [25–29], and its application to auditing built traits is an even smaller fraction of the broader literature [4,5].

In recent years, computer vision models that employ deep learning algorithms have significantly advanced the extraction and analysis of urban environmental features through street-level imagery. Using Convolutional Neural Networks (CNNs), studies have quantified subjective urban perceptions [30,31], safety conditions [32], physical disorder [33,34], greenery levels [18,19], and architectural characteristics at a granular scale [20]. Notably, the Place Pulse dataset pioneered by Salesses, et al. [21] enabled extensive analysis of perceived safety, wealth, and liveliness through crowd-sourced pairwise comparisons of SVI images. Building upon this dataset, subsequent studies such as those by Naik et al. [35,36] expanded methodological frameworks by integrating CNN-based computer vision techniques to quantify and spatially map built change, visual quality, and perceived socioeconomic variables, highlighting the potential of deep learning in urban analytics.

Ilic, et al. [4] coined the term "deep mapping" to describe the application of computer vision and deep learning in the field of gentrification research, where street-level imagery is processed to identify perceptual qualities of the built environment. Deep mapping can be best understood as an automated version or AI simulation of an in-person field survey. Instead of manually recording changes to properties by walking through a neighborhood, the model mimics an auditor's perceptions of visual changes and learns from the patterns of pre-labeled audits.

The research on deep mapping applications for gentrification is limited in number but promising. Ilic, et al. [4] pioneered the application of computer vision for gentrification contexts in their map of built changes for Ottawa, Canada. They applied a Siamese Convolutional Neural Network (SCNN) to learn how to identify gentrification-indicative changes in before-and-after images of a residential property. In constructing a protocol for labeling the training data, the authors are inspired by the criteria set by Hammel and Wyly [11] and Heidkamp and Lucas [37] in their gentrification audits where, for each pairwise image, the authors recorded instances of significant property improvement and investment, such as a loss of a building followed by a new-build development, major renovations, and landscaping. The audit considered changes that are structurally sound and a clear visual improvement. The SCNN model achieved a final test accuracy of 95.6% and an Area Under the Curve (AUC) score of 84%.

Thackway, et al. [5] applied a SCNN model to identify visual gentrification indicators for neighborhoods in Sydney, Australia. The authors leveraged a Residual Network with 50 weight layers (ResNet-50). The ResNet-50 model is pre-trained on ImageNet. ResNet is powerful in its capacity to circumvent the issue of diminishing performance at the deeper layers of the convolutional neural network through the application of skip connections and has been constructed with a variety of layer numbers (e.g., ResNet-18, ResNet-50, ResNet-101). The ResNet-50 model is widely used due to its high levels of performance in image classification tasks. Pairwise images were categorized according to a similar protocol to Ilic, et al. [4] where major upgrades to the frontage and new-build construction are both considered. The SCNN model's mapped output verified the results of a gentrification map based on socioeconomic indicators. The final, fine-tuned ResNet-50 model achieved a reasonably high performance with an 84.8% test accuracy and a 75.6% AUC score.

In our study, we integrate qualitative findings to the development of our deep mapping model. We propose a system for drawing on community-informed insights on gentrification qualities to guide the model training process. The primary aim of the research is to promote a methodology that stipulates the target features underlying gentrification activity and move beyond broad terms like "upgrades" or "improvements" to a more detailed, locale-specific diagnostic of gentrification characteristics. In this way, we can ensure that variations in performance across ML model applications are not due to differing personal perceptions of gentrification across research groups and promote needed discourse on the variable ways in which gentrification alters the built environment. The ML component of our study demonstrates how even opaque, high-level quantitative research can be merged with qualitative discussion to produce internally consistent records of gentrification changes.

## Materials and methods

We select the study area of the City of Philadelphia, Pennsylvania due to recent trends of gentrification. We focus specifically on residential new-build development and exclude commercial, industrial, and mixed land uses. Using Principal Components Analysis (PCA), we develop a composite score to rank tracts according to how likely they are to be gentrifying based on census-based, sociodemographic characteristics. From these filtered tracts, we reach out to a variety of neighborhood groups. The organizations that responded comprise of our focus group selection (e.g., Norris Square, Tacony, Port Richmond).

In all focus group meetings, participants associated gentrification changes with new-build gentrification and developer-led construction rather than to classic gentrification and the gentrifier-led rehabilitation of historic housing stock. These findings are corroborated by journalistic sources, with several articles citing the Riverwards neighborhoods of East Kensington, Northern Liberties, and Fishtown in Northeast Philadelphia as hot spots of new-build gentrification activity [38,39].

We applied a deep mapping SCNN model in the form of ResNet-50 to identify new-build gentrification. The methodology of the research is illustrated in Fig 1.

Before implementing the SCNN model, we applied a novel methodology for developing a protocol to identify gentrification in SVI, as illustrated in Part (A) of Fig 1. First, we implemented a Principal Components Analysis (PCA) to identify neighborhoods that are demonstrating socioeconomic signs of gentrification. Coupled with our regional familiarity with

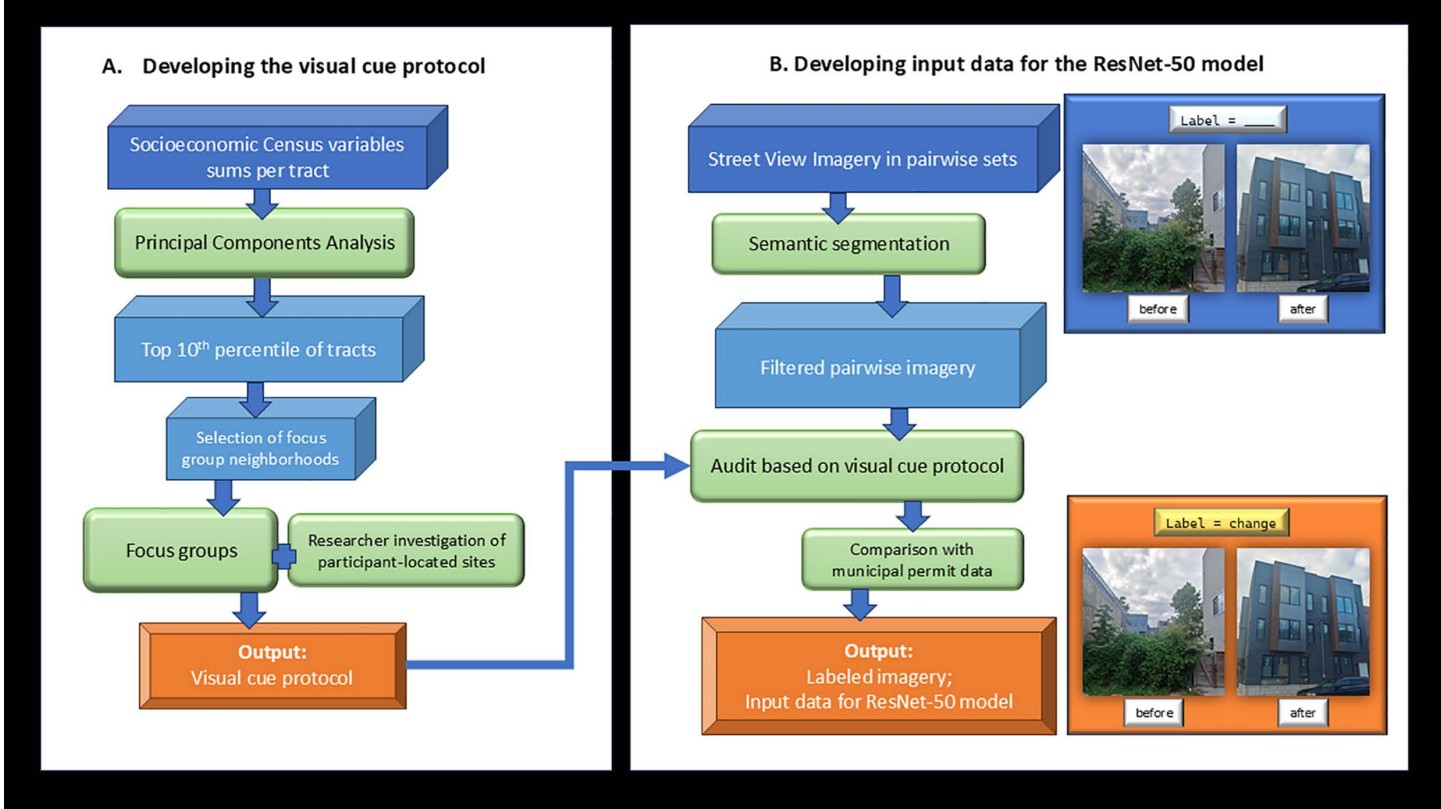

**Fig 1. Flowchart for developing and implementing the image recognition model.** Methodology behind developing a protocol that defines new-build gentrification traits (Part A) and applying the protocol on Street View Imagery (SVI) pairwise imagery of residential frontages (Part B). Photos were taken by the research team and are for illustrative purposes.

the study area, we used this information to identify three target neighborhoods to hold focus groups in (i.e., Tacony, Port Richmond, and Norris Square). These focus groups comprised the qualitative component to our research. We then used the insights of community members to identify the characteristics of a new-build building that are gentrification-indicative (the "Visual cue protocol" in Fig 1).

In Part B of Fig 1, we developed the input data for the deep mapping model ResNet-50 and audited this data according to the visual cue protocol. We then acquired Philadelphia License and Inspection (L&I) data to compare our labeled data with permit records. We ran our SCNN ResNet-50 deep mapping model and assessed the ability of the model to identify cases of development accurately and reliably.

### Principal Components Analysis (PCA)

To identify areas that demonstrate gentrification-indicative levels of socioeconomic shift, we ran a Principal Components Analysis (PCA). PCA can be implemented to produce a composite score by drawing out meaningful information from a wide array of features [40]. The composite score consists of a linear combination of the original Census inputs in a way that maximizes the dataset's variance.

Data is drawn from the American Community Survey (ACS) 5-year estimates for the years 2006–2010 and 2017–2020 [41]. This 7–14-year gap provides a sufficiently long time period for neighborhood change to be discernible in the data and is comparable to other ML-based gentrification studies that choose similar ten-year time ranges

[25,28]. ACS data is drawn at the geographic scale of the Census tract, the smallest unit of measure for independent sampling.

We followed closely the PCA approach applied by previous gentrification studies, Reades, et al. [25] and Owens [42], and integrated the same choice of variables, adapted for US Census data conventions. Four variables are selected for the analysis: median housing price, median household income, educational attainment (percentage of residents with attainment of a bachelor's degree or higher), and white-collar occupational shares (percentage of residents in white-collar occupational classes).

## Development of an auditing protocol with focus group insights

We held qualitative focus groups with residents in three target neighborhoods (i.e., Port Richmond, Tacony, and Norris Square), identified through the PCA step described above, our personal familiarity with the study area, and corroborated by journalistic sources that reported new-build gentrification, to categorize the visual characteristics that describe new build gentrification [43–45]. Our focus group protocol (#31477) was approved by the A1 committee at [redacted for peer review] University on March 21, 2024. The IRB approved the use of verbal informed consent. Before beginning each session, participants were provided with a written consent form and the facilitator reviewed the study procedures and confidentiality protections. Participants then provided verbal consent prior to participation. This consent was documented in the facilitator's field notes and witnessed by a member of the research team present during the session.

From May 15, 2024 to December 27, 2024, we contacted community organizations in each of the neighborhoods to help recruit residents through their local connections and social media channels. Each focus group was held in a neighborhood location (e.g., local library). Two members of the research team led each focus group, with one person posing questions and the other taking notes. Each focus group had four to ten people participating and the discussions lasted about an hour. The majority of participants were middle-aged women that were long-time residents of their neighborhoods, with many having spent decades volunteering to make their neighborhoods more green, liveable places, such as creating community gardens and gathering spaces, and tending to vacant lots to deter crime. Although the participants represent a proportion of the population that are more likely to be invested in local politics and in neighborhood outcomes, thus introducing a level of sampling bias, we posit that the insights of this particular subpopulation are beneficial to the project due to their deep knowledge of constructions, demolitions, and population shifts, ranging back decades. The research team also made note of specific addresses and cross streets where gentrification-associated changes were said to have occurred and corroborated every participant-cited location through historical GSV imagery.

The focus group guide included questions about perceptions of gentrification happening in the neighborhood, signs of gentrification in buildings and along business corridors, and perceptions of whether gentrification had changed the accessibility of places and services in the neighborhood (see appendix for interview protocol).

In developing a protocol to categorize instances of new-build gentrification, we took on a two-part process. First, we analyzed the specific properties that participants recognized as being connected to new-build gentrification. As many of these participants had resided in their neighborhoods for decades, they could provide us with a detailed account of street intersections and landmarks where they had observed development that they deemed gentrification related. By examining these changes via historical GSV imagery, we expanded upon the more general terms used in the focus groups (e.g., "modern," "boxy") with more architecturally specific language whilst ensuring that we were looking at the same target features as the participants. Whereas the final audit was done by sorting through a repository of pairwise images, this initial exploration is performed by manually interacting with historical changes through a 3D virtual walk-through the neighborhood.

Second, we constructed a survey via the Qualtrics interface to ensure that the research team were in consensus on how to categorize images. The Qualtrics survey contained a small random sample of 57 pairwise images from neighborhoods in Northeast Philadelphia that were intensely gentrifying according to our PCA results. Each before- and

after-image was taken a decade apart from the neighborhoods of Tacony, East Kensington, West Kensington, Upper Kensington, Olde Kensington, Richmond, Port Richmond, Fishtown, and Northern Liberties.

After compiling our findings from the focus groups, the manual audit of target areas the focus group participants identified, and the Qualtrics survey, we produced a list of architectural traits and built qualities that were indicative of new-build gentrification (Table 1).

We then utilized this protocol to produce an audit of new-build gentrification cues, as identified in historical SVI (see Fig 2).

As visualized in Fig 2 with illustrative before- and after-imagery, each pair represents a change in a single location over time that is marked as either being indicative of gentrification (label = "change") or not (label = "no change"). The first two pairs represent built indicators of a gentrification change according to our protocol, corresponding to the boxy design, black window paneling, bump out windows that are larger and of different sizes, and a contrasting mix of building materials.

## Data acquisition of Street View Imagery (SVI)

To acquire SVI, we downloaded GSV data via the Google API (https://developers.google.com/maps/documentation/street-view/overview) for panoramas (360° views of the street) of residential frontages in the city of Philadelphia. Residential properties were identified using land use data from *OpenDataPhilly*, a publicly available data catalog for municipal and non-municipal datasets (https://opendataphilly.org/). Using the GIS software QGIS, we extracted centroids from each residential lot to determine the latitude/longitude coordinates used for querying the Street View API.

A buffer of 25 meters was applied around each centroid to ensure comprehensive coverage of residential streetscapes. Using the Google Maps API, we retrieved all available street view image identifiers within these buffers, providing geographic coordinates and image capture dates. We selected images from two distinct time periods: 2009–2013 for the prior image and 2017–2021 for the posterior image. This 5-to-11-year gap ensured that sufficient urban changes could be captured in the dataset. While GSV imagery has been available in the region since 2007, images from 2007–2009 were excluded due to inconsistencies such as blurriness, tilt, and variations in saturation, which could negatively impact model performance.

After acquiring the panoramic images, we applied a perspective transformation to extract four static images from each panorama at fixed angles: 0°, 90°, 180°, and 270°, with a field of view (FOV) set to 120 degrees [46]. These views captured

**Table 1. List of built indicators that are connected to new-build gentrification.**

Presence of a new-build structure with the following traits:
*For all new-build residential buildings:*
Modern or contemporary architectural style
Minimalist design
Three-stories or an increased number of floors compared to older, incumbent buildings
Increased building density
Increased floor-to-area (FAR) ratio
"Boxy" buildings, box-like structure
Use of primary colors and accents
Contrasting mix of building materials with color differences
Homogenous design across rowhomes
Sleek, contemporary font interface for unit number
Privacy fences
Black paneling around windows
Windows that are larger than those of older, incumbent buildings
Contemporary encasement windows
Bump out windows
Windows of different proportions in the same building

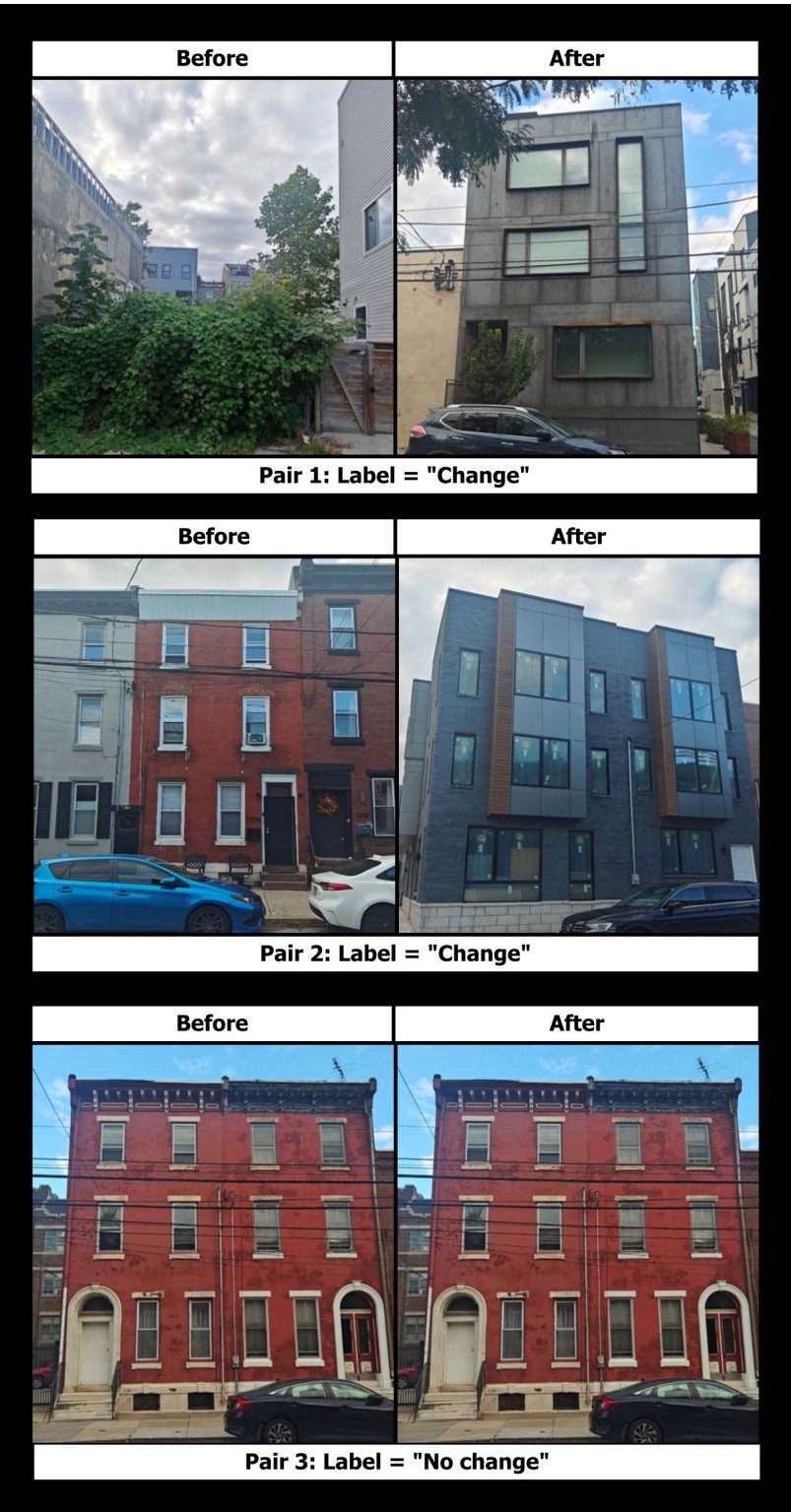

**Fig 2. Training data for image recognition model.** Examples of pairwise imagery in our new-build gentrification audit and associated labels of "No Change" versus "Change" which indicates new-build gentrification is apparent. In order to comply with CC-BY copyright, the photos were taken by the research team for illustrative purposes.

both the streetscape and perpendicular directions, ensuring a comprehensive representation of the built environment. In total, we collected 1,389,059 panoramas, from which static images were extracted and processed for use in the ML model.

We then acquired permit records from *OpenDataPhilly*. The dataset, maintained by the City of Philadelphia's Department of Licenses & Inspections (L&I), contains records of building and zoning permits issued between 2007 and 2023 (https://www.phila.gov/departments/department-of-licenses-and-inspections/). These permit records provided us a method for validating the results of our audited SVI pairs. 712,847 building and zoning permits were issued in the study area over the selected time period.

## Semantic segmentation and data preparation for deep mapping

To accurately track changes in buildings over time, we implemented a spatial, temporal, and semantic filtering approach. First, we applied a spatial and temporal filtering process to identify candidate image pairs. Specifically, we selected images that were captured within a 5-meter radius of each other and ensured that one image was taken between 2009–2013 while the other was from 2017–2021.

Before running the deep mapping model, we filtered out images in the dataset with a higher presence of obstructions that could potentially deter the model from honing in on the target feature of the residential property. We cleaned the data with a semantic segmentation approach using a pre-trained EfficientViT model [47] which identified the relative proportion of certain elements such as vegetation, sky, buildings, roads, and vehicles.

For training the semantic segmentation model, we utilized the *Cityscapes* dataset, a widely recognized benchmark dataset for urban scene understanding (https://www.cityscapes-dataset.com/). *Cityscapes* consists of high-resolution street-level imagery from multiple urban environments, with meticulously annotated pixel-wise labels covering diverse categories, including roads, buildings, vehicles, vegetation, pedestrians, and sky. The EfficientViT model, pre-trained on this dataset, achieved a mean Intersection over Union (mIoU) of 83.2, demonstrating robust segmentation capabilities. By applying this model to our collected imagery, we systematically extracted fine-grained semantic features.

We constrained the proportion of cars to $\leq 0.20$, vegetation to $\leq 0.30$, and the combined proportion of road and sky to $\leq 0.65$. These thresholds minimized mismatches caused by excessive obstructions, such as tree cover or parked vehicles, which could obscure the buildings in the images. After applying these selection criteria, we successfully identified 17,108 valid building image pairs that met all conditions and discarded 1,371,951 pairs with visual obstructions present.

After filtering with semantic segmentation, the 17,108 image pairs were manually audited based on predefined criteria established through the focus group discussion to ensure that they accurately represented gentrification-related changes in Philadelphia. We observed unexpected obstructions in the imagery data that were not filtered out during semantic segmentation, such as SEPTA overhead railway awnings, non-residential buildings on residential sites due to zoning changes, and images where part of the new-build construction was cut off. To account for these obstructions, we manually pared down the dataset to 1,040 pairwise images that unambiguously represented residential, new-build, gentrification-indicating development, discarding a total of 16,068 pairs.

We then partitioned the data to create balanced training and validation sets. Initially, we performed a 20% split to set aside a portion of the dataset specifically for testing. The remaining 80% was then further divided into 70% for training and 30% for validation, ensuring that the model was exposed to a diverse set of urban transformation examples while maintaining sufficient data for evaluation. To ensure a representative distribution of classes, stratified sampling was employed during both partitioning steps. This approach preserved the ratio of gentrification (1) and non-gentrification (0) cases across the different subsets, preventing any class imbalances that could bias the model.

## Deep mapping model

We employed a Siamese Convolutional Neural Network (SCNN) to identify changes indicative of new-build gentrification using the prepared SVI with the binary labels of "change" and "no change." The model architecture applied in this study is a SCNN utilizing a ResNet-50 backbone model trained on ImageNet dataset [48], as shown in Fig 3. The Siamese

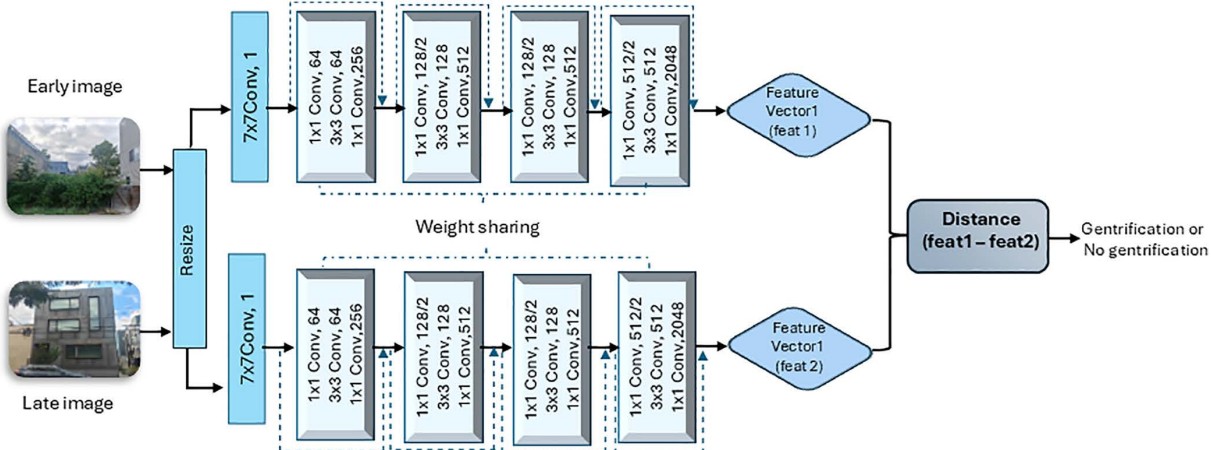

**Fig 3. ResNet-50 Model Architecture Flowchart.** ResNet-50 architecture for new-build gentrification prediction using pairwise Street View Imagery (SVI) as data inputs. Building photos were taken by the research team and are for illustrative purposes.

network was implemented using Python and the PyTorch library. To match the ImageNet dataset on which the backbone ResNet-50 weights, the training images were resized to 224 × 224 pixels, converted to tensors, and normalized using the standard ImageNet mean and standard deviation.

The Siamese ResNet-50 model consisted of 49 convolutional layers with two fully connected layers to focus solely on feature extraction (Fig 3). Both images in each pair were processed through the same ResNet-50 backbone to extract their respective feature vectors. To adapt the model for gentrification detection in Philadelphia, we replaced the original classification head with two fully connected layers. The first fully connected layer reduced the dimensionality of the feature vectors, enabling the model to focus on the most relevant features. This reduction is important in our case because subtle architectural changes were key indicators of gentrification in the SVI dataset. The absolute difference between the two feature vectors was then calculated, highlighting the differences between the images. This difference was passed through the second fully connected layer, a Rectified Linear Unit (ReLU) activation function and a final sigmoid activation function to produce a binary output indicating the likelihood of gentrification (i.e., 1 for "gentrification" and 0 for "no gentrification"). The Siamese network processed both images in parallel, using the same weights for both, ensuring that the model learned the same feature extraction process for both images in the pair.

Optimization of the Siamese network was performed using the Adam optimizer, chosen for its adaptive learning rates and efficient handling of sparse gradient updates. To enhance the robustness and generalizability of the model, extensive data augmentation techniques, including random flips, affine transformations, color jittering, shearing, and rotations, were applied during training. To improve generalization, we applied dropout with a rate of 0.5 to the first fully connected layer (after the ReLU activation) and L2 regularization (weight decay) to penalize large weights. These strategies helped improve the model's ability to handle the variability in architectural styles and image composition across the dataset.

## Results

### Validating audit labels with permit data

To compare the audited data with permit records, we mapped the Kernel Density Estimate (KDE) patterns of both datasets. KDE is a non-parametric statistical technique that produces a smooth, continuous function approximating the distribution of the dataset. The KDE value λ describes how the intensity of points varies in a given locale according to

the probabilistic density function (pdf). We employed a built-in ArcGIS geoprocessing tool to run the KDE analysis with a Gaussian kernel. KDE analyses employ a distance decay function and do not integrate any geographical delineations, but for the sake of discussion we refer to the trends by planning district (e.g., South, West) and by neighborhood boundaries. See S2 Fig for a reference map of planning districts.

From Fig 4, both maps demonstrate development patterns prevalent in Lower North Philadelphia and the Riverwards, South Philadelphia, and Upper Central. The Northeast, Northwest, and the Southwest Philadelphia regions show minimal levels of new-build gentrification-related development patterns.

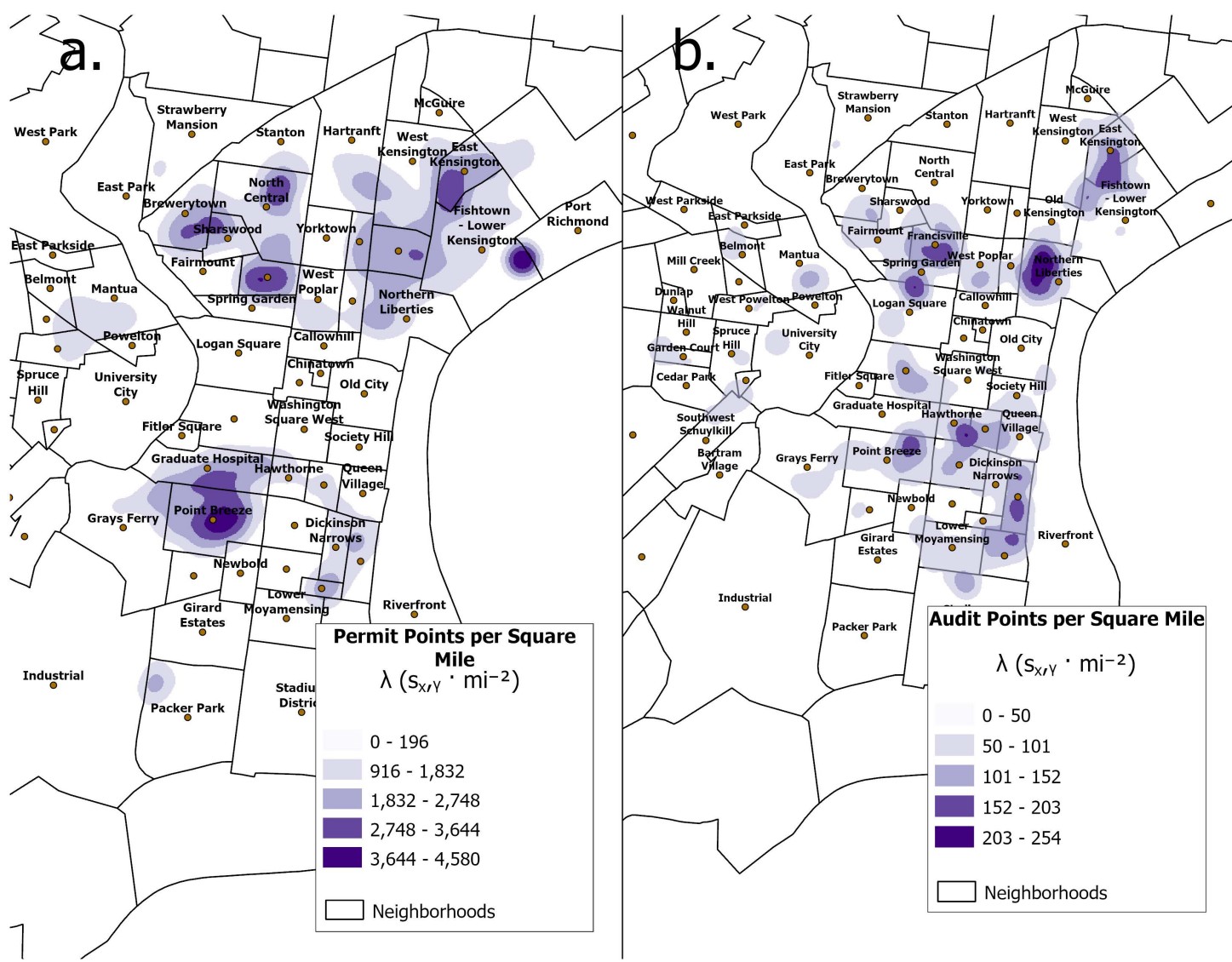

**Fig 4. Comparative new-build development heatmaps.** Comparison between Kernel Density Estimate (KDE) maps of OpenDataPhilly Licenses and Inspections (L&I) Building and Zoning Permits data (a) and audited data points (b). Census tract boundaries and Licenses and Inspections (L&I) Building and Zoning Permits generated using data from Tim Wisniewski (2016), licensed under the MIT License. © Tim Wisniewski. Retrieved from: https://open-dataphilly.org/datasets/census-tracts/ and https://opendataphilly.org/datasets/licenses-and-inspections-building-and-zoning-permits/.

In both maps, the neighborhoods of East Kensington, Francisville, Northern Liberties and Point Breeze are of note as potential new-build gentrification sites. Our audit underestimated the degree of development in upper Brewerytown, North Central, and Old Kensington, neighborhoods which are dense hotspots of new development according to permit data. This discrepancy might be attributed to the sparsity of GSV sample points in these neighborhoods. The lack of workable GSV data could derive from a variety of factors, from a higher frequency of street obstructions (e.g., cars, pedestrians, gates) to lower visibility due to weather conditions. These three neighborhoods underwent significant conversions of formerly industrial land use sites into residential housing, such as the refurbishment of old brewery factories into luxury loft housing in Brewerytown. This type of zoning change may show up on permit data but result in a limited availability of years for GSV, as these areas were once private, gated industrial zones that were inaccessible to GSV image capture vehicles.

The permit data demonstrated notable discrepancies in the lack of development around the area intersecting Spring Garden and Logan Square and in the Rittenhouse neighborhood. We manually inspected the blocks where the permit data lacks records of development, and speculated that a potential reason for this discrepancy could be that fit-outs, which are categorized as renovations in L&I permit data, could be mistaken for new-build development when visually inspecting audit side-by-side audit images. In dense Philadelphia neighborhoods, fit-outs can result in unrecognizable facades depending on the degree to which the structure is reconstructed. In extreme cases, only the original framework and internal wooden beams are preserved, resulting in a built change that eludes a straightforward categorization as a renovation or a new-build construction.

## Deep mapping model performance

The performance of the fine-tuned ResNet-50 model was evaluated using five classification metrics (Table 2). The model reached 84.0% accuracy after introducing a dropout layer with dropout rate of 0.5 into the first fully connected layers and applying L2 regularization to penalize large weights. To assess its ability to distinguish between gentrified locations, we calculated the AUC, which is 84.0%. This suggests the model can reliably separate the two categories of gentrified and non-gentrified. The model also achieved a precision of 0.78 and a recall of 0.88. In other words, it correctly labels 78% of the locations it identifies as "gentrified," and successfully captures 88% of all actual gentrified areas. The resulting F1-score of 0.82 shows a balanced trade-off between precision and recall.

A closer look at the confusion matrix (Fig 5) reveals that the model accurately identified 79 locations without new-build gentrification and 86 locations with new-build gentrification. However, it misclassified 25 non-gentrification locations as new-build gentrification and 8 new-build gentrification locations as non-gentrification. This misclassification pattern suggests a slight bias toward false positives, which aligns with the observed precision-recall trade-off. Overall, these results highlight the potential of deep learning with Siamese networks, particularly when using a ResNet-50 backbone, to effectively capture signals of urban transformation. The high recall is an especially promising metric of reliable model performance, as it ensures the detection of most true gentrification instances in this study.

## Discussion and conclusion

The literature has sparse information on how new-build gentrification manifests in the built landscape [but see 1, 47, 48] and even fewer robust, quantitative inquiries due to limitations in data availability and quality. As the built characteristics of cities are complex and varied, the physicality of gentrification's changes on these city landscapes are also variable, context-dependent, and evolve over time [49]. The current research proposes that this complexity should be

**Table 2. Summary of ResNet-50 Fine-tuning results.**

| Test accuracy | AUC | Precision | Recall | F1-score |
|---|---|---|---|---|
| 0.84 | 0.84 | 0.78 | 0.88 | 0.82 |

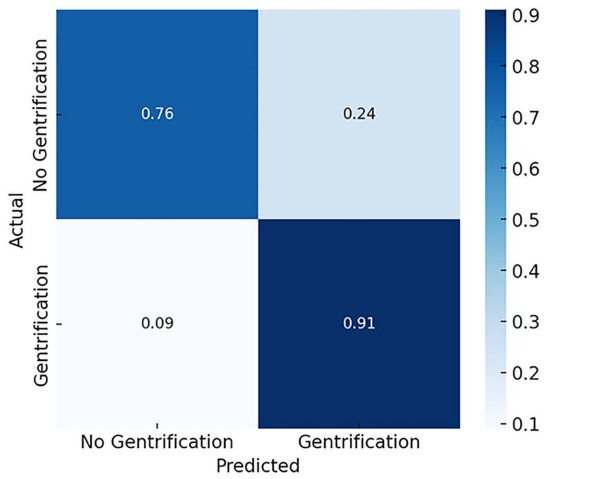
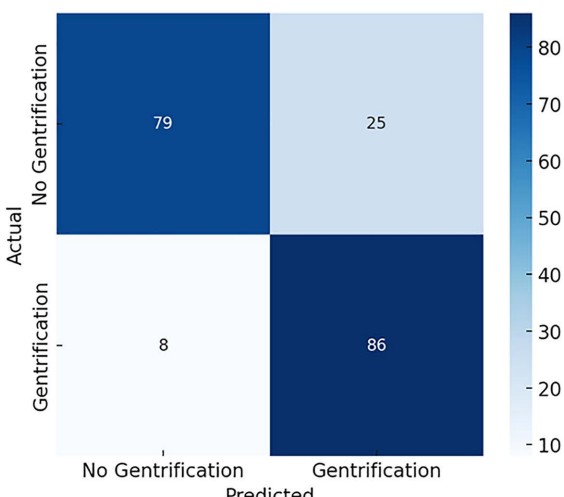

**Fig 5. Classification results on gentrified and non-gentrified paired images.** Comparison of the normalized matrix displaying classification probabilities (left) and the actual confusion matrix with counts of correctly and incorrectly classified cases (right).

accommodated when auditing gentrification's built traits via a mixed-methods approach that integrates community perspectives and historical street view imagery. Whereas prior applications of gentrification deep mapping models are quantitative-centric [4,5], our research takes on a qualitative component that defines the ways in which we train our machine learning model.

We developed and implemented a novel mixed-methods approach to demonstrate how ML can be merged with community-based discussions to produce a more replicable methodology for auditing gentrification changes. Whereas prior studies established that deep mapping can circumvent the need to deal with inconsistent permit data [4], and be combined with sociodemographic metrics to produce maps of gentrification [5], we demonstrate how qualitative research can be integrated to produce a more transparent, reproducible metric for labeling street view imagery (SVI) in a deep mapping context.

We audited 17,108 pairwise images and further pared down the data to 1,040 points that we believe are clear, visual representations of new-build gentrification changes over the time ranges of 2009−2013–2017−2024. By comparing our audit to permit records for new construction, we demonstrated that both datasets demonstrate similar visual trends in KDE maps. The final fine-tuned ResNet-50 model achieved an 84.0% test accuracy and a reasonably high AUC of 84.0%. The results show that deep mapping models can label regionally specific perceptions of new-build gentrification with a reasonably high level of accuracy despite limitations in data availability and the higher prevalence of visual noise.

The ResNet-50 model was sensitive to changes in more than one attached property and cases where the top floors of the new-build property are cut off and partially visible or off-center, suggesting that the ResNet-50 model can identify new-build construction even in instances when the property is not centered. The model was also capable of marking instances with properties that contained a variety of architectural styles and components, such as bump outs, mixes in building materials, and changes in building height (see Table 1 to reference the auditing protocol), although the black-box nature of the algorithm prevents us from saying with certainty whether the model was attending to these localities in the image consistently. For future studies, certain Explainable Artificial Intelligence (XAI) tools could help identify areas of the image that were significant to the model prediction, such as Class Activation Maps [50,51] which are specialized for CNN architectures.

When examining the true negative cases, the model demonstrates the ability to mark pairwise images as "no gentrification" when there are additions to structures that are of the same architectural style as the original building and changes to the color of the façade, demonstrating a noteworthy sensitivity of the model to more complex judgements.

The false positives of the model are largely attributed to instances with vehicle obstructions in terms of differences in the presence of vehicles in a before- and after-image and the presence of pedestrians. We also observe that the blurring out of certain properties for privacy reasons can confuse the model. False negatives can be attributed to images where a new high-rise development is visible far off in the distance, or in cases where the new-build development is the same height as incumbent buildings. This suggests the model prioritizes shifts in building height, specifically when a new building is significantly taller (a characteristic in our visual cue protocol).

The presence of obstructions and consequential lack of workable GSV data prevents the research from producing a detailed map of gentrification-related new-build development in Philadelphia. Moreover, the uneven distribution of the permit dataset limits us from performing a detailed validation of our findings. To the best of our knowledge and at this current time period, there is no way to acquire a well-distributed dataset marking new-build residential development for the Philadelphia region. With a semantic segmentation algorithm, we discarded 1,371,951 image pairs. We then manually parsed through 17,108 image pairs and discarded 16,068 pairs due to the presence of visual obstructions (e.g., the presence of transient objects, trolley bridges) and non-residential land uses (e.g., hospitals, commercial buildings). This built-in limitation brings into question the usability of ML-based image models as a replacement for permit data when it comes to packed urban areas with high levels of noise. However, the results suggest that image recognition models can attain reasonably high performance with limited datasets and may be capable of performing in more visually noisier environments than presented in previous research [4,5]. We propose that such image models may be capable of being applied as replacements or supplements to permit data in cities with dense, attached housing and higher quality street view imagery, even if limits in GSV image quality prevented the research from producing detailed data outputs for our specific study area.

The deep mapping model appears to be competent at identifying subtle changes despite the high presence of noise, the presence of multiple buildings in an image, and the complexity of these street scenes. In sum, the ResNet-50 model appears to be honing in on the architectural traits listed in the protocol, rather than marking every significant change in frontage and every new-build construction as a gentrification positive case.

The small dataset raised concerns about overfitting, as the base ResNet-50 model quickly adapted to limited visual patterns. To mitigate this, we applied visually tested augmentations, horizontal flips, affine shearing, random crops, mild rotations, translations, and tuned brightness and contrast to emulate the variability in Philadelphia's street-view imagery. We also tested three dropout layers, as in prior studies, but found they flattened performance around 70%. A dropout rate of 0.5 on the fully connected layer and a tuned weight decay value of 0.0060 stabilized training and improved generalization. Finally, a *ReduceLROnPlateau* scheduler (factor = 0.3, patience = 2) offered the best balance: larger patience values delayed adaptation, while smaller factors caused early convergence, making this configuration most effective for maintaining steady learning dynamics.

Our model performance metrics are comparable in performance with Thackway, et al. [5] and their research of gentrification in Sydney, Australia. However, the performance of our model is less successful than that of Ilic, et al. [4], the pioneering application of deep mapping for gentrification modeling, which achieves the highest accuracy at 95.6% and an AUC of 84% for their study area neighborhoods in Ottawa, Canada. Ilic, et al. [4] focus on single-family, detached, and semi-detached residential homes, whereas our pairwise images incorporate dense, urban areas of Philadelphia with attached rowhomes, apartment complexes, and townhome structures. As camera locations can have slightly shifted field of views (FOVs) in before and after images, this can result in a different number of rowhomes being present in each pairwise set and result in false positive detections of new-build development.

The high performance of the model, despite the limitations in the size of the data, could be attributed to several factors. For one, unlike Thackway, et al. [5] and Ilic, et al. [4], we examined new-build gentrification and solely focused on cases of new-build development and the addition of floors, whereas former research included renovations and new landscaping. Although there is a dearth of literature on the variations of gentrification's impacts on the architectural composition of cities, we posit that Philadelphia's form of new-build gentrification development is highly visually distinct. In focus groups, there was no hesitation when asked to describe what gentrification looked like to them, and unanimous consensus in every focus group that the new buildings "stuck out like a sore thumb" compared to the incumbent buildings. This visual distinctive nature could be a major reason the model could perform with high accuracy from a small set of carefully curated examples. Lastly, we posit that the preparatory steps taken prior to auditing could benefit the model's performance. We propose that an initial survey amongst the auditing team could be useful for future audit-based research, as researchers within the team were initially not in consensus with the labeling of more complex street scenes, such as the presence of a new-build development that is partially visible in the background, or the nuances of distinguishing between a fit-out and new-build construction.

Our research shows that Philadelphia possesses a distinct type of new-build gentrification architectural design. Although the traits of a new-build gentrification building are similar across Philadelphia, the impacts on communities vary greatly from neighborhood-to-neighborhood in terms of the types of valued amenities that are lost in construction and the sites/land usage where new-build development arises.

Finally, we have shown that ethnographic accounts are valuable for producing replicable, transparent audits of gentrification phenomena in application to machine learning contexts. With more reliable methods and data on gentrification's effect on the built environment, urban planners can gain insight into how certain types of development result in inequitable effects and organizations can identify neighborhoods that require protection from displacement [52,53]. Additionally, communities and stakeholders can apply this research to promote media awareness and gain support from policy makers [53,54].

## Supporting information

**S1 Fig. Principal Components Analysis (PCA) map.** Principal Components Analysis (PCA) output indicating gentrification-indicative socioeconomic change. Census tract boundaries generated using data from Tim Wisniewski (2016), licensed under the MIT License. © Tim Wisniewski. Source: https://opendataphilly.org/datasets/census-tracts/. Basemap data © OpenStreetMap contributors, licensed under the Open Data Commons Open Database License (ODbL). Source: https://www.openstreetmap.org.
(TIF)

**S2 Fig. Planning district map.** Reference map for Philadelphia's neighborhood boundaries. Adapted from Philadelphia Neighborhoods by Robert Cheetham (2014). © Robert Cheetham, licensed under **CC BY 4.0**. Source: https://opendata-philly.org/datasets/philadelphia-neighborhoods/.
(TIFF)

**S1 Table. Principal Components Analysis (PCA) summary statistics for 2010.** PCA output as importance of PC components for the year 2010 ACS Census variables.
(DOCX)

**S2 Table. Principal Components Analysis (PCA) summary statistics for 2021.** PCA output as importance of PC components for the year 2021 ACS Census variables.
(DOCX)

## Acknowledgments

We wish to acknowledge the contributions of our research assistants from Temple University, Department of Geography, Environment and Urban Studies: Camryn Stuhlmuller, Coby Lindenmuth, Desiree Sustello, Wesley Martin, and Zoe Longley.

## Author contributions

**Conceptualization:** Maya Mueller, Hamil Pearsall, Simi Hoque.

**Data curation:** Maya Mueller, Isaac Quaye, Shengao Yi, James Foley, Reeya Shah.

**Formal analysis:** Maya Mueller, Isaac Quaye, Shengao Yi, James Foley, Reeya Shah.

**Funding acquisition:** Hamil Pearsall, Simi Hoque.

**Investigation:** Maya Mueller, Isaac Quaye, Shengao Yi, James Foley, Reeya Shah.

**Methodology:** Maya Mueller, Isaac Quaye, Shengao Yi, James Foley, Reeya Shah, Hamil Pearsall.

**Project administration:** Hamil Pearsall, Simi Hoque.

**Resources:** Xiaojiang Li, Hamil Pearsall, Simi Hoque.

**Software:** Xiaojiang Li.

**Supervision:** Xiaojiang Li, Hamil Pearsall, Simi Hoque.

**Validation:** Maya Mueller, Isaac Quaye.

**Visualization:** Maya Mueller, Isaac Quaye, James Foley, Reeya Shah.

**Writing – original draft:** Maya Mueller, Isaac Quaye, Shengao Yi, James Foley, Reeya Shah, Hamil Pearsall.

**Writing – review & editing:** Maya Mueller, Hamil Pearsall, Simi Hoque.

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
