## [Decision Letter · Decision Letter 0]

2 Sep 2025

Dear Dr. Mueller,

We look forward to receiving your revised manuscript.

Kind regards,

Floris Vermeulen

Academic Editor

PLOS ONE

Journal Requirements:

2. In the ethics statement in the Methods, you have specified that verbal consent was obtained. Please provide additional details regarding how this consent was documented and witnessed, and state whether this was approved by the IRB

3. Please note that PLOS One has specific guidelines on code sharing for submissions in which author-generated code underpins the findings in the manuscript. In these cases, we expect all author-generated code to be made available without restrictions upon publication of the work. Please review our guidelines at https://journals.plos.org/plosone/s/materials-and-software-sharing#loc-sharing-code and ensure that your code is shared in a way that follows best practice and facilitates reproducibility and reuse.

4. We note that your Data Availability Statement is currently as follows: All relevant data are within the manuscript and in Supporting Information files.

6. We note that Figure 4 and S2 in your submission contain map/satellite images which may be copyrighted. All PLOS content is published under the Creative Commons Attribution License (CC BY 4.0), which means that the manuscript, images, and Supporting Information files will be freely available online, and any third party is permitted to access, download, copy, distribute, and use these materials in any way, even commercially, with proper attribution. For these reasons, we cannot publish previously copyrighted maps or satellite images created using proprietary data, such as Google software (Google Maps, Street View, and Earth). For more information, see our copyright guidelines: http://journals.plos.org/plosone/s/licenses-and-copyright.

1. You may seek permission from the original copyright holder of Figure 4 and S2 to publish the content specifically under the CC BY 4.0 license.

7. We note that Figures 1-3, 6 and S1 in your submission contain copyrighted images. All PLOS content is published under the Creative Commons Attribution License (CC BY 4.0), which means that the manuscript, images, and Supporting Information files will be freely available online, and any third party is permitted to access, download, copy, distribute, and use these materials in any way, even commercially, with proper attribution. For more information, see our copyright guidelines: http://journals.plos.org/plosone/s/licenses-and-copyright.

1. You may seek permission from the original copyright holder of Figures 1-3, 6 and S1to publish the content specifically under the CC BY 4.0 license.

8. We notice that your supplementary figures are uploaded with the file type 'Figure'. Please amend the file type to 'Supporting Information'. Please ensure that each Supporting Information file has a legend listed in the manuscript after the references list.

Reviewers' comments:

Reviewer's Responses to Questions

**Comments to the Author**

1. Is the manuscript technically sound, and do the data support the conclusions?

Reviewer #1: Yes

Reviewer #2: Yes

2. Has the statistical analysis been performed appropriately and rigorously?

Reviewer #1: Yes

Reviewer #2: Yes

3. Have the authors made all data underlying the findings in their manuscript fully available?

Reviewer #1: No

Reviewer #2: Yes

4. Is the manuscript presented in an intelligible fashion and written in standard English?

Reviewer #1: Yes

Reviewer #2: Yes

Reviewer #1: The authors present a mixed-methods framework to detect new-build gentrification using a deep learning model trained on community-informed visual cues from street-level imagery. The paper is well-written and the results are promising.

After filtering and manual auditing, only 1,040 image pairs were retained for model training. Although the authors argue this is sufficient due to the distinct visual features of gentrification in Philadelphia, the small sample raises concerns about overfitting and the model’s ability to generalize.

Reviewer #2: Thank you for the opportunity to review the article entitled “Developing a machine learning model to map new-build gentrification: A mixed-methods approach”. The article introduces a mixed-methods procedure to measure the locally specific characteristics of new-build gentrification. The authors focus on Philadelphia and identify gentrification traits that are characteristic to the city, but also emphasize that variation at the neighborhood level exists when it comes to the impact of new-build gentrification.

The combination of Machine Learning and Focus Groups that the authors introduce to identify gentrification traits is very interesting and the methods have been applied thoroughly. It remains a bit unclear why the authors focus on new-build gentrification. On lines 170-171 it is argued that new-build gentrification is central to the paper because the concept was emphasized in focus groups, but on lines 221-224 the authors state that the focus groups have been selected based on Principal Components Analysis and journalistic sources that report new-build gentrification. If you select the focus groups (partially) on new-build gentrification, it may be no surprise that participants emphasize new-build gentrification. The question is why the authors used new-build gentrification as a condition for selecting the focus groups? I can imagine why it is interesting to focus on new-build gentrification, but the authors can make this more explicit in their paper. It would also be good to address the difference between new-build and traditional gentrification in the abstract or introduction of the article, so the reader knows from the start what new-build gentrification is and why the article focuses on this concept.

With regard to the focus groups, the authors could share more information on the participants of the focus groups (for instance median age, level of education, male/female). As the composition of the focus groups may impact the gentrification traits that are identified. This could especially be interesting if you would replicate this study in other localities and find very different gentrification traits. Are these differences then due to local characteristics or due the differences in the composition of the focus groups?

**Do you want your identity to be public for this peer review?** For information about this choice, including consent withdrawal, please see our Privacy Policy

Reviewer #1: No

Reviewer #2: No

---

## [Author Response · Author response to Decision Letter 1]

16 Dec 2025

All responses to the editor are uploaded as a separate word document file entitled "Response to the editor.docx" and the responses to the reviewers have been addressed in the "Response to the reviewer.docx" file.

The responses to the reviewers are as follows:

Reviewer #1:

The authors present a mixed-methods framework to detect new-build gentrification using a deep learning model trained on community-informed visual cues from street-level imagery. The paper is well-written and the results are promising.

After filtering and manual auditing, only 1,040 image pairs were retained for model training. Although the authors argue this is sufficient due to the distinct visual features of gentrification in Philadelphia, the small sample raises concerns about overfitting and the model’s ability to generalize.

Response to Reviewer #1:

Thank you for taking the time to read through and comment on the manuscript. Your critique addresses a major limitation of the research and one of the primary hurdles the research team faced when developing the SCNN image recognition model. All line numbers are referencing the manuscript with track changes on.

The small sample size wasn’t a methodological choice, but rather due to a lack of quality GSV imagery. Philadelphia is more dense and visually noisy than the regions chosen in prior studies (e.g., select neighborhoods in Ottawa, Canada with single-family detached homes in Ilic et al., 2019). GSV snapshots were frequently obscured by cars, trolley bridges, and other visual obstructions that would render the images unusable. Through semantic segmentation, we ended up discarding this 1,371,951 of images, resulting in 17,108 image pairs for the team to manually label, and the research team further cut down the dataset by this 16,068 during our audit process for the cases where the semantic segmentation algorithm did not pick up on other visual obstructions.

This built-in limitation brings into question the usability of ML-based image models as a replacement for permit data when it comes to packed urban areas and attached rowhome housing, but we found the research to be valuable in the discovery that image recognition models can perform well with limited datasets and noisy imagery. The team is currently working to ascertain whether the model was focusing on target features (e.g., architectural aspects of the building rather than surrounding objects) and examining how to fine-tune image recognition models to function with limited data availability.

Again, the limit in usable GSV sampling points means that, to the best of our knowledge, there is no way to acquire a well-distributed dataset of new-build development for the Philadelphia region. The model output dataset, similar to the permit dataset, is scarce and sparsely distributed. However, we have ascertained that image recognition models can attain reasonably high performance with our limited data, and that these models may be capable of performing with more visually noisier environments than previously thought. We hope our research contributes to an idea of how much these models can be “pushed” for instances where there is higher quality GSV imagery in cities with dense, attached housing.

Rather than offering a replacement for permit data in our selected study area, we reframed the research to promote a novel methodology that combines qualitative, focus group insights in order to create more transparency in how we’re training perceptual image recognition models, further demonstrating the extent to which these models can perform in dense cities like Philadelphia (research contributions are summarized in lines 107-118 in the “Introduction” section), and to demonstrate how to construct and fine-tune these models to improve performance on smaller sized datasets. We agree that the manuscript would be improved upon with a deeper discussion into how limited data availability affects the usability of the research. We have done so in the “Discussion and Conclusion” section, lines 568-584. We also address how we mitigated the possibility of overfitting on lines 590-599.

Reviewer #2:

Thank you for the opportunity to review the article entitled “Developing a machine learning model to map new-build gentrification: A mixed-methods approach”. The article introduces a mixed-methods procedure to measure the locally specific characteristics of new-build gentrification. The authors focus on Philadelphia and identify gentrification traits that are characteristic to the city, but also emphasize that variation at the neighborhood level exists when it comes to the impact of new-build gentrification.

The combination of Machine Learning and Focus Groups that the authors introduce to identify gentrification traits is very interesting and the methods have been applied thoroughly. It remains a bit unclear why the authors focus on new-build gentrification. On lines 170-171 it is argued that new-build gentrification is central to the paper because the concept was emphasized in focus groups, but on lines 221-224 the authors state that the focus groups have been selected based on Principal Components Analysis and journalistic sources that report new-build gentrification. If you select the focus groups (partially) on new-build gentrification, it may be no surprise that participants emphasize new-build gentrification. The question is why the authors used new-build gentrification as a condition for selecting the focus groups? I can imagine why it is interesting to focus on new-build gentrification, but the authors can make this more explicit in their paper. It would also be good to address the difference between new-build and traditional gentrification in the abstract or introduction of the article, so the reader knows from the start what new-build gentrification is and why the article focuses on this concept.

With regard to the focus groups, the authors could share more information on the participants of the focus groups (for instance median age, level of education, male/female). As the composition of the focus groups may impact the gentrification traits that are identified. This could especially be interesting if you would replicate this study in other localities and find very different gentrification traits. Are these differences then due to local characteristics or due the differences in the composition of the focus groups?

Response to Reviewer #2:

Thank you for your deep read of the manuscript and for providing a thoughtful review. All line numbers are referencing the manuscript with track changes on. We address the comments below:

New-build gentrification was a re-occurring theme in the focus groups, with participants focusing solely on new-build developments they connected with gentrification rather than renovations that would be classic gentrification-specific. The process of picking participants and neighborhoods wasn’t a linear process. Rather, we ran a Principal Components Analysis, reached out to a wide variety of neighborhood groups from these tracts, and then further complemented the qualitative analysis with research into journalistic narratives of gentrification in these areas of Philadelphia. From these journalistic sources, we noted that the accounts of new-build, developer-led gentrification were corroborated. We agree that the manuscript could be improved upon by being clearer about this aspect of the methodology and would benefit from additional explanation on the difference between traditional vs. new-build gentrification. These changes can be viewed in the “Introduction”, lines 98-105, and the “Materials and Methods” section, lines 193-204.

Although we originally considered integrating renovations in our auditing protocol, we realized that it was challenging to distinguish between an incumbent, resident-led renovation versus a developer-led, gentrification renovation via GSV pairwise imagery. Renovations can be internal to the building, and even external differences like repainted frontages may be too subtle for our machine learning model. As focus group participants also had no observations regarding gentrification-related renovations, the research team decided to focus solely on new-build gentrification moving forward. We add additional commentary on this aspect of the methodology on lines 103-105.

In regards to the second comment, we did not gather specific information on the demographics of our participants. However, we did observe that participants were overwhelmingly middle-aged women. In Port Richmond and Tacony, there were only women participants. In Norris Square, there were two male participants and two female participants. Participants were long-time residents of their neighborhoods, and several had spent decades volunteering and investing time to make their neighborhoods more green, liveable places (e.g., creating community gardens and gathering spaces, tending to vacant lots to deter crime). Many were invested in neighborhood outcomes and were involved in local politics in attempts to make their neighborhoods safer. However, all participants were against gentrification as a solution to promote capital inflows, and saw these improvements as exclusionary and side-stepping the needs of long-time residents in lieu of marketing property to newcomers.

The profile of the study participants can be a source of bias as residents were not randomly selected, but the insights of this particular subpopulation was beneficial due to their deep, encyclopedic knowledge of the constructions, demolitions, and population shifts, ranging back decades. We were able to make note of specific addresses and cross streets where gentrification-associated changes occurred and corroborated every participant-cited location through historical GSV imagery.

This level of spatial and temporal detail allowed the research team to identify specific, notable developments and study the visual patterns of these buildings. This may have not been possible with a younger demographic of residents who had only been in the neighborhood a few years, or with those who were less socially active in their local communities. Although the participants do not represent a randomly selected sample, they are likely the most observant archivers of these historical changes to the landscape, could recount when these physical changes coincided with demographic shifts (i.e., signifying gentrification), and could provide us with a level of locational detail that was later triangulated with GSV audits.

With your comments in mind, we have added additional discussion on the composition of the focus groups, how we corroborated findings via GSV, and on how our selection of focus groups could be a source of sampling bias in the “Development of an auditing protocol with focus group insights” section, lines 265-275.

The responses to the editor are as follows:

Thank you to the editor for taking the time to evaluate our manuscript. We address the comments below:

https://journals.plos.org/plosone/s/file?id=ba62/PLOSOne_formatting_sample_title_authors_affiliations.pdfa

We ensured that the PLOS ONE style requirements were met, including those for file naming. We were able to find specific guidelines to naming the supplementary materials files and have revised those names accordingly. We could not find specific information about naming the other main submission files, so please let us know if there are any errors on that end.

2. In the ethics statement in the Methods, you have specified that verbal consent was obtained. Please provide additional details regarding how this consent was documented and witnessed, and state whether this was approved by the IRB.

We specify how consent was documented and witnessed on lines 287-289 in the section “Development of an auditing protocol with focus group insights”. Details about IRB approval are provided on lines 276-278. These line numbers reference the manuscript version with track changes on.

3. Please note that PLOS One has specific guidelines on code sharing for submissions in which author-generated code underpins the findings in the manuscript. In these cases, we expect all author-generated code to be made available without restrictions upon publication of the work. Please review our guidelines at https://journals.plos.org/plosone/s/materials-and-software-sharing#loc-sharing-code and ensure that your code is shared in a way that follows best practice and facilitates reproducibility and reuse.

Thank you for your comment. We have shared the minimal dataset via GitHub. The link to the data repository is as follows: https://github.com/niiquaye70/Mapping-New-Build-Gentrification-with-Machine-Learning/tree/main/data. The DOI is https://doi.org/10.5281/zenodo.17352508.

4. We note that your Data Availability Statement is currently as follows: All relevant data are within the manuscript and in Supporting Information files.

Thank you for your comment. We have shared the minimal dataset via GitHub. The link to the data repository is as follows: https://github.com/niiquaye70/Mapping-New-Build-Gentrification-with-Machine-Learning/tree/main/data. The DOI is https://doi.org/10.5281/zenodo.17352508. We have acquired an open-source MIT license as follows:

MIT License

Copyright (c) 2023 NSF #2312047 and #2312048 Projects

Permission is hereby granted, free of charge, to any person obtaining a copy of this software and associated documentation files (the "Software"), to deal in the Software without restriction, including without limitation the rights to use, copy, modify, merge, publish, distribute, sublicense, and/or sell copies of the Software, and to permit persons to whom the Software is furnished to do so, subject to the following conditions:

The above copyright notice and this permission notice shall be included in all copies or substantial portions of the Software.

THE SOFTWARE IS PROVIDED "AS IS", WITHOUT WARRANTY OF ANY KIND, EXPRESS OR IMPLIED, INCLUDING BUT NOT LIMITED TO THE WARRANTIES OF MERCHANTABILITY, FITNESS FOR A PARTICULAR PURPOSE AND NONINFRINGEMENT. IN NO EVENT SHALL THE AUTHORS OR COPYRIGHT HOLDERS BE LIABLE FOR ANY CLAIM, DAMAGES OR OTHER LIABILITY, WHETHER IN AN ACTION OF CONTRACT, TORT OR OTHERWISE, ARISING FROM, OUT OF OR IN CONNECTION WITH THE SOFTWARE OR THE USE OR OTHER DEALINGS IN THE SOFTWARE.

We decline to share the focus group data because it contains sensitive lo

---

## [Editor Report · Decision Letter 1]

13 Jan 2026

Developing a machine learning model to map new-build gentrification: A mixed-methods approach

PONE-D-25-31456R1

Dear Dr. Mueller,

We’re pleased to inform you that your manuscript has been judged scientifically suitable for publication and will be formally accepted for publication once it meets all outstanding technical requirements.

Kind regards,

Floris Vermeulen

Academic Editor

PLOS One
---

## [Editor Report · Acceptance letter]

PONE-D-25-31456R1

PLOS One

Dear Dr. Mueller,

I'm pleased to inform you that your manuscript has been deemed suitable for publication in PLOS One. Congratulations! Your manuscript is now being handed over to our production team.

Kind regards,

on behalf of

Dr. Floris Vermeulen

Academic Editor

PLOS One